# Agree to Disagree: Adaptive Ensemble Knowledge Distillation in Gradient Space

**Shangchen Du**[1*]  **Shan You**[2,3*†]  **Xiaojie Li**[2]  **Jianlong Wu**[4,5†],
**Fei Wang**[2]  **Chen Qian**[2]  **Changshui Zhang**[3,6,7]
[1]School of EECS, Peking University   [2]SenseTime
[3]Department of Automation, Tsinghua University
[4]School of Computer Science and Technology, Shandong University   [5]Zhejiang Laboratory
[6]Institute for Artificial Intelligence, Tsinghua University (THUAI)
[7]Beijing National Research Center for Information Science and Technology (BNRist)
dushangchen@pku.edu.cn, {youshan,lixiaojie,wangfei,qianchen}@sensetime.com,
jlwu1992@sdu.edu.cn, zcs@mail.tsinghua.edu.cn

## Abstract

Distilling knowledge from an ensemble of teacher models is expected to have a more promising performance than that from a single one. Current methods mainly adopt a vanilla average rule, *i.e.*, to simply take the average of all teacher losses for training the student network. However, this approach treats teachers equally and ignores the diversity among them. When conflicts or competitions exist among teachers, which is common, the inner compromise might hurt the distillation performance. In this paper, we examine the diversity of teacher models in the gradient space and regard the ensemble knowledge distillation as a multi-objective optimization problem so that we can determine a better optimization direction for the training of student network. Besides, we also introduce a tolerance parameter to accommodate disagreement among teachers. In this way, our method can be seen as a dynamic weighting method for each teacher in the ensemble. Extensive experiments validate the effectiveness of our method for both logits-based and feature-based cases.

## 1  Introduction

Recently, deploying deep neural networks on edge devices has appealed much attention due to their achieved success in various tasks [32, 19, 36, 34, 13, 35, 33, 15, 38]. For the sake of the limited computational budget, light yet promising networks are thus favored. With this aim, efforts have been made to design more compact networks, compress or prune a pretrained model [28, 29] or even search for a tiny backbone network [37, 39]. Inspired by the teacher-student paradigm [17, 40, 42] in human education, knowledge distillation (KD) [9] serves as a complementary method to further boost the performance of a small student network by distilling the knowledge from a large teacher network. Concretely, (softened) logits of the student network are usually encouraged to mimic those of the teacher network. Besides, some approaches also implement distillation on the intermediate features via various manners, such as hints [22], distribution of features [10] and local correlations [14].

Two heads are better than one. Instead of leveraging a single teacher network, distilling knowledge from an ensemble of teacher networks is supposed to achieve more promising performance. In real life, students can learn better following multiple teachers, which is also true for ensemble learning

---

[*]Equal contributions. Work was done during Du's internship at SenseTime.
[†]Corresponding authors.

and KD. Many approaches have investigated the performance improvement in ensemble KD. For example, [41] leveraged averaged softened outputs of multiple teachers and relative dissimilarity between intermediate representations of different examples. [24] used perturbed logits to simulate learning from multiple teachers while [6] argued to randomly select a teacher to strengthen the complementariness in the ensemble. Besides, [21] distilled the distribution of predictions from the ensemble, rather than the averaged output, into a single model.

However, most existing ensemble KD methods simply adopt a vanilla manner for distilling the knowledge or information in ensembles, *i.e.*, averaging the KD losses or averaging softened outputs of all teachers (AVER), which we find equivalent mathematically. However, this vanilla average method neglects the diversity in the ensemble, and there might be conflicts, competition or even noise among all teachers. Inspired by the human education, we can analyse this ensemble KD from the gradient space. During the learning process of the student, teachers can provide the learning target (objective function) as well as the direction of the study. Likewise, in the training of the student network, gradients provided by teacher networks can serve as learning directions. In the ensemble KD, every teacher will provide a gradient which is expected to help the student to converge most quickly. Nevertheless, for the vanilla average, the optimizing direction might be determined by some dominant teachers, and can not take the full advantage of the knowledge within the ensemble.

In this paper, we propose a new adaptive ensemble knowledge distillation (AE-KD) method to encourage the comprehensive distillation from an ensemble. Concretely, to handle the inner conflicts, we formulate ensemble KD as a multi-objective optimization (MOO) problem [26] and use multiple-gradient descent algorithm (MGDA) to probe a Pareto optimal solution that accommodates all teachers as much as possible. Considering there might be noisy and weak teachers in the ensemble, we introduce a tolerance parameter to control the disagreement among teachers. In this way, the optimizing direction will be less influenced by those stray or noisy teachers, which deteriorates the performance accordingly. Our method can be seen as a dynamic weighting strategy for each teacher, so that we can distill knowledge adaptively during the training of the student network. Moreover, our method has friendly interpretability. For example, when it comes to the logits-based KD, our AE-KD can be regarded as a prior alignment of logits between student and all teachers.

Our main contributions can be summarized as follows:

- We revisit ensemble KD from gradient space and formulate it as a multi-objective optimization problem, so that we can better distill knowledge from all the teacher models.
- We introduce a tolerance parameter to control the disagreement among teachers with the aim to accommodate some noisy or weak teachers within the ensemble.
- Our proposed method AE-KD can be regarded as a dynamic weighting strategy to adaptively distill from all teachers.
- We conduct extensive experiments on three benchmark datasets, and results demonstrate the superiority of our method.

## 2 Knowledge Distillation from Ensemble

We first formally introduce the KD method, then we illustrate how the vanilla ensemble KD method functions, including both logits-based and feature-based cases. Given a teacher and a student network, we denote the logits of two networks as $\boldsymbol{a}^t$ and $\boldsymbol{a}^s$. Then KD encourages that the logits of the student network mimic those of the teacher network by minimizing the following loss:

$$\mathcal{L}_{kd} = \mathcal{H}(\boldsymbol{p}^s, \boldsymbol{p}^t) = \mathcal{H}(\sigma(\boldsymbol{a}^s; T), \sigma(\boldsymbol{a}^t; T)) = -\sum_{k=1}^{K} p^t[k] \log p^s[k] = -\langle \boldsymbol{p}^t, \log \boldsymbol{p}^s \rangle, \quad (1)$$

where

- $T$ is the temperature to soften the logits for more fine-grained information,
- $\sigma(\cdot)$ is the softmax operation with temperature $T$,
- $\mathcal{H}(\cdot, \cdot)$ is the cross-entropy loss to measure the discrepancy of softened probabilistic output between the student and teacher,

- $p[k]$ is the $k$-th component of vector $\boldsymbol{p}$,
- $\langle \cdot, \cdot \rangle$ is the inner product of two vectors (tensors).

For the case with ensemble (multiple teacher networks), the vanilla KD loss evolves to match the mean softened output $\boldsymbol{p}^t$ of all teachers. Suppose the ensemble size is $M$. Eq. (1) is then rewritten as:

$$\mathcal{L}_{mkd} = \mathcal{H}(\boldsymbol{p}^s, \frac{1}{M}\sum_{m=1}^{M}\boldsymbol{p}_m^t) = -\langle \frac{1}{M}\sum_{m=1}^{M}\boldsymbol{p}_m^t, \log \boldsymbol{p}^s \rangle = \frac{1}{M}\sum_{m=1}^{M}\mathcal{H}(\boldsymbol{p}^s, \boldsymbol{p}_m^t). \qquad (2)$$

From Eq. (2), we can see that distilling with the averaged softened output of all teachers is actually equal to that with averaged KD loss of Eq. (1) for each teacher. Then we have the following Proposition 1.

**Proposition 1.** *For ensemble knowledge distillation, averaged softened outputs are equivalent to the averaged losses as Eq. (2) since the cross-entropy loss $\mathcal{H}(\cdot, \cdot)$ in Eq. (1) is linear to teacher's output.*

Therefore, we simply use *averaged KD loss* to indicate the vanilla ensemble KD method. Besides the logits-based methods, many approaches also investigate distilling knowledge on the intermediate representations (or features) to further boost the performance. Let $\boldsymbol{f}^t$ and $\boldsymbol{f}^s$ denote the feature maps of the teacher and the student network, respectively. Then the objective of feature-based KD can be generally written as

$$\mathcal{L}_{fkd} = \mathcal{D}(r^t(\boldsymbol{f}^t), r^s(\boldsymbol{f}^s)), \qquad (3)$$

where $r^t$ and $r^s$ are regressor functions or the transformers to align the sizes of feature maps in two networks. $\mathcal{D}(\cdot, \cdot)$ is the distance metric measuring the (relative) similarity of two features. Similarly, for ensemble case, this loss can be formulated as

$$\mathcal{L}_{mfkd} = \frac{1}{M}\sum_{m=1}^{M}\mathcal{D}(r_m^t(\boldsymbol{f}^t), r_m^s(\boldsymbol{f}^s)). \qquad (4)$$

Different types of distance metrics and regressors correspond to different feature-based KD methods. In this paper, we leverage the hints in FitNets [22] for simplicity yet our method applies to other methods similarly. In detail, FitNets uses the squared $\ell_2$ norm as distance metric, then Eqs. (3) and (4) can be written as

$$\mathcal{L}_{fkd} = \frac{1}{2}\|\boldsymbol{f}^t - r(\boldsymbol{f}^s)\|^2, \qquad (5)$$

$$\mathcal{L}_{mfkd} = \frac{1}{2M}\sum_{m=1}^{M}\|\boldsymbol{f}_m^t - r_m(\boldsymbol{f}^s)\|^2, \qquad (6)$$

where $r$ or $r_m$ is the regressor function on top of the output feature map of the student.

As a result, the final optimization objective of ensemble knowledge distillation is to minimize the following loss function

$$\mathcal{L}_{ens} = \mathcal{H}(\boldsymbol{y}, \sigma(\boldsymbol{a}^s; 1)) + \lambda \cdot \mathcal{L}_{mkd} + \beta \cdot \mathcal{L}_{mfkd}, \qquad (7)$$

where $\boldsymbol{y}$ is the ground-truth label vector, and $\lambda$ and $\beta$ are tunable parameters to balance different kinds of losses. Note that when $\beta$ equals to 0, Eq. (7) refers to the logits-based KD.

## 3 Our Proposed Approach

### 3.1 Rethinking Ensemble KD in Gradient Space

As shown in Eqs. (2) and (3), we can see that the vanilla average strategy simply takes into account of all teacher losses equally. The final update direction from ensemble guidance is the average of all gradients in terms of all teacher losses. However, since teachers may provide different learning directions (*i.e.*, gradients) for the student, there might be conflicts and competitions among teachers. As a result, the final learning direction will be determined by the most dominant teacher, thus guidance from other teachers is weakened. From the perspective of gradient space, to address this dominance issue and distill from ensemble more comprehensively, our intuition is that we can observe the

guidance of all teachers as much as possible. Inspired by the success of multi-objective optimization (MOO) problem [5, 25, 4, 20], we formulate the ensemble KD as a MOO problem with each objective corresponding to each teacher, which has been applied to recommender system [18], multi-task learning [26, 16], and multi-agent reinforcement learning [31].

Concretely, when distilling from the ensemble as Eq. (7), besides the gradient of the supervision from ground-truth labels, we also want to determine a direction that comes from all teachers in the ensemble. Suppose the $\ell_m^t(\boldsymbol{\theta})$ is the KD loss regarding the $m$-th teacher and $\boldsymbol{\theta}$ is the parameter of the student network. Then the direction $\boldsymbol{d}$ should accommodate as more teachers as possible, *i.e.*,

$$\mathbf{L} = (\langle \nabla_{\boldsymbol{\theta}} \ell_1^t(\boldsymbol{\theta}), \boldsymbol{d} \rangle, ...., \langle \nabla_{\boldsymbol{\theta}} \ell_m^t(\boldsymbol{\theta}), \boldsymbol{d} \rangle, ..., \langle \nabla_{\boldsymbol{\theta}} \ell_M^t(\boldsymbol{\theta}), \boldsymbol{d} \rangle) \in \mathbb{R}^{M-}, \tag{8}$$

which means we encourage each $\langle \nabla_{\boldsymbol{\theta}} \ell_m^t(\boldsymbol{\theta}), \boldsymbol{d} \rangle$ to be non-positive so that each teacher loss can get decreased. Obtaining such a proper direction can resort to gradient-based MOO methods, such as multiple-gradient descent algorithm (MGDA) [4]. However, when the diversity of ensemble is significant or the conflicts and noise among teachers are severe, gradients of all teachers do not always reach an agreement, making it hard for the student to choose which direction to follow. Averaging teachers' opinions is a compromise but fails to grasp the informative knowledge within the ensemble.

### 3.2 Agree to Disagree: Incorporating Controllable Tolerance for Ensemble KD

As previously illustrated, when dealing with the ensemble KD, we hope for a descent direction that can accommodate all teacher losses, so that it is more robust to conflicts or competitions among teachers. However, conservatively accepting the directions from all teachers is also not a good option, since the diversity of teachers could be significant and there might be some weak or noisy teachers mingled in the ensemble. In this way, when distilling knowledge from the ensemble, we need to incorporate the *disagreement* into the determination of the descent direction. Namely, we encourage that teachers agree to disagree, and the student takes supervision from those actually helpful teachers.

Moreover, when optimizing the student network using stochastic gradient descent (SGD) optimizers, different teachers may have their preferences over different mini-batches. Following [26, 16], we formulate the learning of the target descent direction as the following optimization problem:

$$\min_{\boldsymbol{d}, v, \xi_m} v + C \cdot \sum_{m=1}^{M} \xi_m + \frac{1}{2} \|\boldsymbol{d}\|^2, \text{s.t. } \langle \nabla_{\boldsymbol{\theta}} \ell_m^t(\boldsymbol{\theta}^{(\tau)}), \boldsymbol{d} \rangle \leq v + \xi_m, \ \xi_m \geq 0, \ \forall m \in [1:M], \quad (9)$$

where

- $\boldsymbol{\theta}^{(\tau)}$ is the parameter of the student network at iteration $\tau$,
- $\ell_m^t(\boldsymbol{\theta}^{(\tau)})$ is the KD loss regarding the student and the $m$-th teacher,
- $\boldsymbol{d}$ is the optimizing direction that we hope to learn,
- $v$ is the magnitude that reflects whether $\boldsymbol{d}$ is a decreasing direction for loss $\ell_m^t(\boldsymbol{\theta}^{(\tau)})$,
- $C > 0$ is the regularization parameter,
- $\xi_m$ is the slack variable that allows the violation of $\langle \nabla_{\boldsymbol{\theta}} \ell_m^t(\boldsymbol{\theta}^{(\tau)}), \boldsymbol{d} \rangle \leq v$.

Like Eq. (8), Problem in Eq. (9) aims to probe a descent direction that can decrease all the teacher losses. However, we introduce a slack variable for each teacher loss, so that the model can adaptively determine which teacher(s) will violate the inequality constraint, and might not get decreased eventually. By dint of the Lagrange multipliers, we can have the dual problem of Eq. (9) as

$$\min_{\boldsymbol{\alpha}} \frac{1}{2} \left\| \sum_{m=1}^{M} \alpha_m \nabla_{\boldsymbol{\theta}} \ell_m^t(\boldsymbol{\theta}^{(\tau)}) \right\|^2, \text{s.t. } \sum_{m=1}^{M} \alpha_m = 1, \ 0 \leq \alpha_m \leq C, \ \forall m \in [1:M]. \quad (10)$$

Moreover, by examining the KKT conditions, we can have the solution of Eq. (9), *i.e.*, descent direction $\boldsymbol{d}^{(\tau)}$

$$\boldsymbol{d}^{(\tau)} = -\sum_{m=1}^{M} \alpha_m^* \nabla_{\boldsymbol{\theta}} \ell_m^t(\boldsymbol{\theta}^{(\tau)}), \tag{11}$$

where $\boldsymbol{\alpha}^*$ is the solution of the dual problem Eq. (10). Besides, we have the following Theorem 1 (see supplementary material for proof).

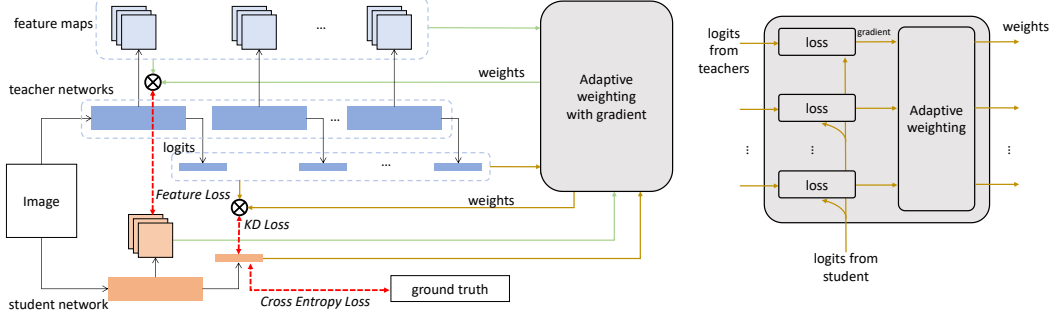

Figure 1: Framework of our AE-KD. (Left) The complete architecture of our proposed method. Logits (and feature maps) from teachers and the student are collected to compute teacher-specific weights. Next, loss from each teacher-student pair is combined with the weights and then we perform regular training with backpropagation. (Right) Illustration for weight computation process using logits. Each logits $a_m^t$ from teachers computes loss with the logits $a^s$ from the student. The gradients for $a^s$ are sent to compute the weights.

**Theorem 1.** *Suppose the solution of Eq. (9) is $(\boldsymbol{d}^{(\tau)}, v^{(\tau)}, \xi_m^{(\tau)})$, then the following holds:*

1. *If $\boldsymbol{\theta}^{(\tau)}$ is Pareto critical, then $\boldsymbol{d} = \boldsymbol{0}$.*

2. *If $\boldsymbol{\theta}^{(\tau)}$ is not Pareto critical, then for $m = 1, ..., M$,*

$$\langle \nabla_{\boldsymbol{\theta}} \ell_m^t(\boldsymbol{\theta}^{(\tau)}), \boldsymbol{d}^{(\tau)} \rangle \leq v^* + \xi_m^* = -\left\| \boldsymbol{d}^{(\tau)} \right\|^2 + \xi_m^* - C \sum_{m=1}^{M} \xi_m^*. \tag{12}$$

*$\boldsymbol{d}^{(\tau)}$ will be a descent direction that at least decreases for the losses satisfying $\xi_m^* - C \sum_{m=1}^{M} \xi_m^* \leq 0$.*

From Eqs. (12) and (10), we can see $C \in [1/M, 1]$ acts as a constant controlling the tolerance of disagreement among teachers. With the decrease of $C$, more tolerance of disagreement among the gradients is allowed. If $C = 1/M$, according to the KKT condition of problem Eq. (10), we have $\alpha_m = 1/M$ for all $m \in [1 : M]$, which means that we simply leverage the average of teacher gradients regardless of any disagreement in gradient space. In contrast, with $C = 1$ we impose no hard box constraint on the value of $\alpha_m$. [26] demonstrates that the obtained gradient direction will accommodate all teachers, and no tolerance of disagreement is allowed. Concretely, three cases can be listed as follows:

- $C = 1$: Each teacher loss gets decreased; however, it implies that some noisy or weak teachers also get decreased, which is not necessary.

- $C = 1/M$: Not every teacher loss gets decreased; however, each teacher is equally involved in the objective.

- $C \in (1/M, 1)$: Most teacher losses get decreased; and teachers are involved adaptively, which is more robust.

However, calculating the gradient over parameters $\boldsymbol{\theta}^{(\tau)}$ can be fairly time-consuming. Following [26], we turn to its upper bound, *i.e.*,

$$\min_{\boldsymbol{\alpha}} \frac{1}{2} \left\| \sum_{m=1}^{M} \alpha_m \nabla_{\boldsymbol{Z}} \ell_m^t(\boldsymbol{\theta}^{(\tau)}) \right\|^2, \text{s.t.} \sum_{m=1}^{M} \alpha_m = 1, \ 0 \leq \alpha_m \leq C, \ \forall m \in [1 : M], \tag{13}$$

where $\boldsymbol{Z}$ is the shared feature map over all teachers. Besides, if we assume $\frac{\partial \boldsymbol{Z}}{\partial \boldsymbol{\theta}^{(\tau)}}$ is full-rank, then solving Eq. (13) still ensures Theorem 1 according to the Theorem 1 in [26]. Eq. (13) is a typical One-class SVM problem with constraint $0 \leq \alpha_m \leq C$, and can be easily solved by LIBSVM [1] or other off-the-shelf solvers. Our method (called AE-KD) is illustrated in Figure 1.

### 3.3 Case Study: AE-KD as an Adaptive Weighting Strategy

Now we investigate how our formulation functions in two major KD categories, *i.e.* logits-based KD and feature-based KD.

#### 3.3.1 Logits-based KD

For vanilla KD, the loss for each teacher amounts to Eq. (1), and all teachers provide guidance on the output of the student network. In this way, the losses of all teachers fully share the parameters on the student network. For our method, the shared feature map amounts to the logits $a$ in the output layer. As [7], the gradient of vanilla KD loss Eq. (1) w.r.t. logits $a$ can be calculated as

$$\nabla_{\boldsymbol{a}} \mathcal{L}_{vkd} = \nabla_{\boldsymbol{a}} \mathcal{H}(\boldsymbol{p}^s, \boldsymbol{p}^t) = \frac{1}{T}(\boldsymbol{p}^s - \boldsymbol{p}^t). \tag{14}$$

As a result, our formulation for the multiple teachers Eq. (13) will be rewritten as

$$\min_{\boldsymbol{\alpha}} \ \frac{1}{2T^2} \left\| \boldsymbol{p}^s - \sum_{m=1}^{M} \alpha_m \boldsymbol{p}_m^t \right\|^2, \text{s.t.} \ \sum_{m=1}^{M} \alpha_m = 1, \ 0 \le \alpha_m \le C, \ \forall m \in [1:M]. \tag{15}$$

In this sense, our method can be viewed as a *dynamic weighting strategy*. The optimal weight is determined by *prior alignment* of softened probabilistic outputs between the student and all teachers. On the capped simplex (*i.e.* the hard constraint of Eq. (15)), all teachers first mutually align their softened outputs towards that of the student, then they follow the same alignment direction on the losses to fulfill the gradient descent so that part of their losses ($C < 1$) or all of their losses ($C = 1$) can be exactly decreased by the student network. In this way, the guidance from the ensemble is more tolerant to the potential teacher noise or some extremely weak teachers.

From this interpretability, assigning fixed weights might introduce manual bias or preference among teachers, and induce sub-optimal results. Some work such as OKDDip [2] also attempts to leverage self-attention mechanism to compute weights in online KD manner. However, since no disagreement among teachers is considered, the KD performance also has the risk of being distracted by some dominant or weak teachers.

#### 3.3.2 Feature-based KD: Mimicking Feature Maps

In feature-based KD paradigm, parameters used to compute the selected feature map are shared but each regressor for feature alignment is private to each teacher-student pair. Denote the shared feature map as $\boldsymbol{Z}$. For simplicity, we assume the resolution of feature maps between student and teachers are the same in Eq. (3). Then the ensemble KD in Eq. (13) will be written as

$$\min_{\boldsymbol{\alpha}} \frac{1}{2} \left\| \boldsymbol{Z}^s - \sum_{m=1}^{M} \alpha_m \boldsymbol{Z}_m^t \right\|^2, \text{s.t.} \ \sum_{m=1}^{M} \alpha_m = 1, \ 0 \le \alpha_m \le C, \ \forall m \in [1:M]. \tag{16}$$

Similar to the logits-based case, in feature-based KD, teachers also are encouraged to align their feature maps towards that of the student. Then all teachers will reach an agreement despite their disagreement under the control of parameter $C$.

## 4 Experimental Results

In this section, we conduct extensive experiments to demonstrate the effectiveness of our method. We compare our methods on the logits-based and feature-based setting with other commonly used ensemble learning methods. CIFAR10 [12], CIFAR100 [11] and ImageNet [3] are used to evaluate the performance. Furthermore, we evaluate different teacher-student pairs with similar, different and identical architectures to analyze the robustness of our method. Finally, we perform ablation studies to evaluate the effect of parameters in our method. [3]

Table 1: Classification accuracy (%) of student network on CIFAR10 and CIFAR100 with resnet56 teachers. # indicates the ensemble size of teachers. "*" refers to feature-based ensemble KD using AE-KD method. ↑ means the performance improvement of AE-KD with respect to the baseline AVER. Ens refers to the "Ensemble", and we use majority voting to calculate the performance of ensemble.

| | CIFAR10 | | | | | | | | | |
|---|---|---|---|---|---|---|---|---|---|---|
| | resnet20 (91.7) | | | | | MobileNetV2 (75.97) | | | | |
| # | Ens | AVER | AE-KD | FitNets | AE-KD* | Ens | AVER | AE-KD | FitNets | AE-KD* |
| 1 | 93.94 | 91.84 | 91.84 | 91.86 | 91.86 | 93.94 | 75.77 | 75.77 | 77.07 | 77.07 |
| 5 | 95.27 | 91.94 | 92.50 (↑0.56) | 91.96 | 92.58 (↑0.62) | 95.27 | 76.11 | 77.32 (↑1.21) | 78.19 | 81.16(↑2.97) |
| 10 | 95.58 | 91.96 | 92.54 (↑0.58) | 92.09 | 92.67 (↑0.58) | 95.58 | 77.47 | 79.27 (↑1.80) | 78.30 | 81.83(↑3.53) |
| 15 | 95.75 | 92.01 | 92.55 (↑0.54) | 92.18 | 92.89 (↑0.71) | 95.75 | 77.96 | 80.17 (↑2.21) | 78.51 | 82.14(↑3.63) |
| 20 | 95.74 | 92.22 | 92.70 (↑0.48) | 92.31 | 92.93 (↑0.62) | 95.74 | 78.89 | 80.41 (↑1.52) | 78.94 | 82.24(↑3.30) |
| 25 | 95.78 | 92.50 | 92.74 (↑0.24) | 92.50 | 93.01 (↑0.51) | 95.78 | 79.25 | 80.70 (↑1.45) | 80.60 | 82.56(↑1.96) |
| | CIFAR100 | | | | | | | | | |
| | resnet20 (69.06) | | | | | MobileNetV2 (64.60) | | | | |
| # | Ens | AVER | AE-KD | FitNets | AE-KD* | Ens | AVER | AE-KD | FitNets | AE-KD* |
| 1 | 72.41 | 70.49 | 70.49 | 70.55 | 70.55 | 72.41 | 66.44 | 66.44 | 66.84 | 66.84 |
| 5 | 77.42 | 70.78 | 71.37 (↑0.59) | 70.97 | 71.95 (↑0.98) | 77.42 | 66.82 | 67.27 (↑0.45) | 66.98 | 67.63(↑0.65) |
| 10 | 79.15 | 70.84 | 71.40 (↑0.56) | 71.26 | 71.99 (↑0.73) | 79.15 | 66.90 | 67.47(↑0.57) | 67.05 | 67.71 (↑0.66) |
| 15 | 79.55 | 71.21 | 71.48 (↑0.27) | 71.49 | 72.12 (↑0.63) | 79.55 | 67.03 | 67.59 (↑0.56) | 67.13 | 67.92 (↑0.79) |
| 20 | 79.83 | 71.47 | 71.66 (↑0.19) | 71.58 | 72.21 (↑0.63) | 79.83 | 67.09 | 67.84 (↑0.75) | 67.17 | 68.14 (↑0.97) |
| 25 | 80.01 | 71.57 | 71.84 (↑0.27) | 71.70 | 72.36 (↑0.66) | 80.01 | 67.19 | 67.93 (↑0.74) | 67.45 | 68.20 (↑0.75) |

## 4.1 Results on the CIFAR10 and the CIFAR100 Datasets

CIFAR10 [12] consists of 50K training images and 10K test images from 10 classes while CIFAR100 [11] has the same amount of images but from 100 classes. We use resnet56 [8] as the teacher network and train 25 teacher models on both datasets for 240 epochs with the learning rate starting from 0.05 and multiplied by 0.1 at 150, 180, 210 epochs. We investigate two teacher-student pairs with the same (resnet56-resnet20 [8]) and different (resnet56-MobileNetV2 [23]) architectural types.[4] For MobileNetV2, we use the same training strategy as teachers except that the initial learning rate is 0.01. For resnet20, we train for 350 epochs with the learning rate starting from 0.05 and divide it by 10 every 50 epochs since the 150th epoch. $\lambda$ in Eq.(7) is set to 0.9 while $\beta$ is determined via cross-validation from $\{10^{-1}, 1, 10, 100, 1000\}$. The temperature in Eq. (1) is set to 4.

Table 1 presents the classification accuracy of ensemble KD with different ensemble sizes using the baseline AVER and our AE-KD method. From the results, we can observe that AE-KD outperforms AVER for all ensemble sizes in both logits-based and feature-based settings, verifying the effectiveness of our method. Besides, when the student has a different architectural type (MobileNetV2) from teachers (resnet56), our method still enjoys the superiority, which indicates the robustness to network types of our method.

Table 2: Classification accuracy (%) of student networks with the same architecture as teachers on CIFAR10 and CIFAR100. Symbols are the same with Table 1.

| | CIFAR10 | | | | | CIFAR100 | | | | |
|---|---|---|---|---|---|---|---|---|---|---|
| # | Ens | AVER | AE-KD | FitNets | AE-KD* | Ens | AVER | AE-KD | FitNets | AE-KD* |
| 1 | 93.94 | 92.58 | 92.58 | 92.49 | 92.49 | 72.41 | 73.58 | 73.58 | 73.61 | 73.61 |
| 5 | 95.27 | 92.67 | 92.88 (↑0.21) | 92.64 | 93.47 (↑0.83) | 77.42 | 74.39 | 74.88 (↑0.49) | 74.58 | 75.02(↑0.44) |
| 10 | 95.58 | 92.82 | 93.23 (↑0.41) | 92.67 | 93.64 (↑0.97) | 79.15 | 74.66 | 74.96 (↑0.30) | 74.69 | 75.16(↑0.47) |
| 15 | 95.75 | 92.89 | 93.71 (↑0.82) | 92.77 | 93.66 (↑0.89) | 79.55 | 74.74 | 75.14 (↑0.40) | 74.75 | 75.37(↑0.62) |
| 20 | 95.74 | 92.87 | 93.75 (↑0.88) | 92.90 | 93.76 (↑0.86) | 79.83 | 74.92 | 75.45 (↑0.53) | 75.04 | 75.53(↑0.49) |
| 25 | 95.78 | 93.73 | 94.20 (↑0.47) | 93.13 | 93.88 (↑0.75) | 80.01 | 75.04 | 75.62 (↑0.58) | 75.10 | 75.69(↑0.59) |

In addition to the "big-teacher and small-student" setting of KD, we also implement experiments on the student with identical architecture with teachers. In detail, we use resnet56 as both the student and teachers. From Table 2, we can see that although the performance gap between teachers and students is not significant, our method can still achieve better performance than the baseline AVER method by adaptively weighting the advantage of each teacher.

Moreover, previous ensemble consists of teacher networks with the same architectural type. We also investigate the KD from an ensemble of various architectures. We select resnet110, WRN-40-

Table 3: Accuracy (%) on ImageNet of ResNet18 student and ResNet50 teacher networks. Symbols are the same with Table 1.

| | Ensemble | | AVER | | AE-KD | |
|---|---|---|---|---|---|---|
| # | Top-1 | Top-5 | Top-1 | Top-5 | Top-1 | Top-5 |
| 1 | 75.67 | 92.50 | 67.81 | 88.21 | 67.81 | 88.21 |
| 3 | 76.85 | 93.60 | 67.85 | 88.39 | 68.28 (↑0.43) | 88.21 (↓0.18) |
| 5 | 77.52 | 93.85 | 68.18 | 88.47 | 69.14 (↑0.96) | 88.93 (↑0.46) |

2 [43] and vgg13 [27] as teacher networks with accuracy 74.31%, 75.61% and 74.64% classification accuracy on CIFAR100 dataset, respectively, and we implement ensemble KD on the resnet20 student network. Results show that our AE-KD can achieve 70.16% accuracy while our baseline method AVER only has 69.40%, showing our superiority of dealing ensemble KD even with various architectures.

## 4.2 Results on the ImageNet Dataset

ImageNet [3] contains 1.2M images from 1K classes for training and 50K for validation. For models trained on ImageNet, we follow the setting in [21]. We choose ResNet50 as teacher networks and ResNet18 as the student network. We compare the Top-1 and Top-5 classification accuracy on the validation dataset and summarize the results in Table 3. Compared with AVER, we can see that in the challenging large-scale ImageNet dataset, our method still has the superiority over the baseline AVER method. For example, with ensemble size 5, our method improves about 0.96%/0.46% on Top-1/Top-5 accuracy. This validates the effectiveness of our method when it comes to large-scale datasets in practice.

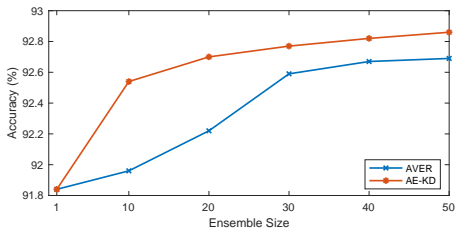

Figure 2: Accuracy on CIFAR10 with different ensemble sizes.

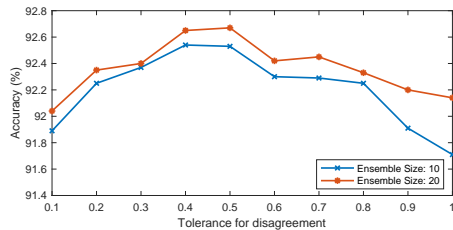

Figure 3: Accuracy on CIFAR10 with different level of tolerance for disagreement.

## 4.3 Ablation Studies

**Effect of ensemble size.** To investigate the effect of the ensemble size on KD, we train the resnet20 student network on the CIFAR10 dataset with different ensemble sizes of resnet56 teacher networks. As shown in Figure 2, increasing ensemble size exerts a positive effect on the performance. With more teachers involved, the proportion of weak teachers will tend to be steady. Thus for AVER, their influence on the performance will gradually reduce to a certain level, and the accuracy will increase with more teachers at first, but stabilize afterwards. Nevertheless, our AE-KD still works consistently better than AVER with a comparable and stable margin for large ensemble sizes.

**Effect of tolerance for disagreement.** Tolerance of disagreement among teachers decides how much teachers compromise and reach consensus. We evaluate the impact of tolerance using different values for disagreement parameter $C$. The experimental setting is similar to that above, and the results are presented in Figure 3. It can be seen that AE-KD works better for $C \in (1/M, 1)$. This might result from that teachers are involved more adaptively and good guidance from teachers are strengthened, thus the student is trained in a better direction. The optimal value for $C$ varies according to different experimental settings. A well-selected disagreement parameter (*e.g.*, 0.5 for ensemble size 20 in Figure 3) can keep teachers in balance and boost the accuracy of the student.

# 5    Conclusion

We propose a new method for ensemble knowledge distillation named adaptive ensemble knowledge distillation (AE-KD), which utilizes the idea of multi-objective optimization and finds a better learning direction for the student network in every mini-batch. To accommodate the guidance from all teachers, our method uses the gradients of teachers to decide the final direction in a dynamic manner. Our method can be deployed in both logits-based and feature-based ensemble KD, and have friendly interpretability. Extensive experimental results validate the superiority of our method to the baseline averaged rule. For future work, we intend to investigate how to adjust the disagreement parameter dynamically as well.

## Broader Impact

Knowledge distillation (KD) serves as a general technique for boosting training neural networks. And it is more highlighted in the training of smaller networks, which enjoys the prospects to be deployed in various edge devices, such as smartphones, wearable watches and AR glasses. KD enables us to take the full advantage of deep learning power, and introduce it on the edge computation. It is thus promising to realize a truly-intelligent society.

## Acknowledgments and Disclosure of Funding

Shan You is supported by Beijing Postdoctoral Research Foundation. Jianlong Wu is supported by the National Natural Science Foundation of China (grant no. 62006140), the Fundamental Research Funds and the Future Talents Research Funds of Shandong University.

## Footnotes

[3]Code is released on `https://github.com/AnTuo1998/AE-KD`.

[4]We use resnet-d to represent CIFAR-style resnet, ResNet-d to represent ImageNet-style ResNet with BottleNeck blocks, WRN-d-w to represnet wide resnet with depth $d$ and width factor $w$ as in [30].

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
