[Supplementary Material]

# Agree to Disagree: Adaptive Ensemble Knowledge Distillation in Gradient Space (Supplementary Materials)

## A   Proof of Theorem 1

Note that the primal problem Eq. (9) is

$$\min_{\boldsymbol{d},v,\xi_m} v + C \cdot \sum_{m=1}^{M} \xi_m + \frac{1}{2} \|\boldsymbol{d}\|^2, \text{ s.t. } \langle \nabla_{\boldsymbol{\theta}} \ell_m^t(\boldsymbol{\theta}^{(\tau)}), \boldsymbol{d} \rangle \leq v + \xi_m, \ \xi_m \geq 0, \ \forall m \in [1:M]. \quad (17)$$

Its Lagrange function can be written as

$$\mathcal{L}(\boldsymbol{d},v,\xi_m,\alpha_m,\beta_m) = v + C \cdot \sum_{m=1}^{M} \xi_m + \frac{1}{2} \|\boldsymbol{d}\|^2 + \sum_{m=1}^{M} \alpha_m (\langle \nabla_{\boldsymbol{\theta}} \ell_m^t(\boldsymbol{\theta}^{(\tau)}), \boldsymbol{d} \rangle - v - \xi_m) - \sum_{m=1}^{M} \beta_m \xi_m, \quad (18)$$

where $\alpha_m$ and $\beta_m$ are Lagrange multipliers. Then we have

$$\frac{\partial \mathcal{L}}{\partial \boldsymbol{d}} = d + \sum_{m=1}^{M} \alpha_m \nabla_{\boldsymbol{\theta}} \ell_m^t(\boldsymbol{\theta}^{(\tau)}) = 0, \quad \rightarrow \quad \boldsymbol{d} = -\sum_{m=1}^{M} \alpha_m \nabla_{\boldsymbol{\theta}} \ell_m^t(\boldsymbol{\theta}^{(\tau)}), \quad (19)$$

$$\frac{\partial \mathcal{L}}{\partial v} = 1 - \sum_{m=1}^{M} \alpha_m = 0, \quad \rightarrow \quad \sum_{m=1}^{M} \alpha_m = 1, \quad (20)$$

$$\frac{\partial \mathcal{L}}{\partial \xi_m} = C - \alpha_m - \beta_m = 0, \quad \rightarrow \quad \alpha_m + \beta_m = C. \quad (21)$$

The dual problem is thus

$$\min_{\boldsymbol{\alpha}} \frac{1}{2} \left\| \sum_{m=1}^{M} \alpha_m \nabla_{\boldsymbol{\theta}} \ell_m^t(\boldsymbol{\theta}^{(\tau)}) \right\|^2, \text{ s.t. } \sum_{m=1}^{M} \alpha_m = 1, \ 0 \leq \alpha_m \leq C, \ \forall m \in [1:M]. \quad (22)$$

Denote the optimal solution of Eq. (9) and Eq. (10) as $(\boldsymbol{d}^*, v^*, \xi_m^*)$ and $(\alpha_m^*, \beta_m^*)$, respectively. According to KKT condition, we have

$$\alpha_m^*(\langle \nabla_{\boldsymbol{\theta}} \ell_m^t(\boldsymbol{\theta}^{(\tau)}), \boldsymbol{d}^* \rangle - v^* - \xi_m^*) = 0, \quad \beta_m^* \xi_m^* = 0, \quad \alpha_m^* \xi_m^* = C \xi_m^*. \quad (23)$$

In this way,

- if $\boldsymbol{d}^* = \boldsymbol{0}$, then we have $\langle \nabla_{\boldsymbol{\theta}} \ell_m^t(\boldsymbol{\theta}^{(\tau)}), \boldsymbol{d}^* \rangle = 0$, which is a trivial case and corresponds to the Pareto critical point.
- if $\boldsymbol{d}^* \neq \boldsymbol{0}$, we have

$$-\|\boldsymbol{d}\|^2 - v^* - C \sum_{m=1}^{M} \xi_m^* = 0, \quad (24)$$

  which implies that

$$\langle \nabla_{\boldsymbol{\theta}} \ell_m^t(\boldsymbol{\theta}^{(\tau)}), \boldsymbol{d}^* \rangle \leq v^* + \xi_m^* = -\|\boldsymbol{d}\|^2 + \xi_m^* - C \sum_{m=1}^{M} \xi_m^*. \quad (25)$$

Then the proof is completed.