[Reviews · NeurIPS 2020]

Review 1

Summary and Contributions: This paper mainly investigates the knowledge distillation (KD) from an ensemble, which is composed of multiple teacher networks. To fulfill the ensemble KD, most methods simply follow the average rule and neglect the diversity or differences among teachers. In contrast, this paper breaks this routine, and proposes to adaptively weight each teacher in the ensemble so that the knowledge can be leveraged more flexibly. Based on the perspective of multi-objective optimization, this paper introduces a slack variable to model the disagreement among teachers and proposes a gradient descent method accordingly. The proposed method is well explained when it comes to the typical KD which amounts to “logits alignment” for the integration of ensemble. Extensive experimental results on CIFAR10/100 and ImageNet datasets validate the effectiveness of the proposed method.

Strengths: 1.The intuition of this paper is good. Instead of following the routine average rule, this paper proposes to adaptively weight the teachers in the ensemble, and proposes a gradient-based algorithm which can well model the disagreement among teachers. 2.The proposed method has a nice explanation when it comes to both logit-based and feature-based knowledge distillation. The logits alignment perspective seem interesting to me. 3. Elaborate and corroborative mathematical proof are provided to verify the correctness of our algorithm. 4. Extensive experimental results on CIFAR10/100 and ImageNet datasets validate the effectiveness of the proposed method.

Weaknesses: More analysis over the disagreement parameter is encouraged to be given to improve the comprehensiveness of the ablation studies. more references can be included to provide an ampler background for KD.

Correctness: Yes

Clarity: Yes

Relation to Prior Work: Yes

Reproducibility: Yes

Additional Feedback: Please handle the comments accordingly. Post-rebuttal I have read the responses and other reviewers' comments. The authors have addressed most cocerns, I'm satisfied to accept this paper.


Review 2

Summary and Contributions: This paper concentrates on ensemble learning of teachers for better transferring knowledge to student. The authors regard ensemble knowledge distillation as a multi-objective optimization problem that obtains a set of weights for different teachers based on gradients to dominate the learning procedure of the student. Both logit-based and feature-based KD are considered in experiments to explore the applicability of the proposed method.

Strengths: 1. The authors mathematically analysis the equivalent between averaging the KD losses and averaging the softened outputs in AVER. 2. The ensemble knowledge distillation is formulated as a multi-objective optimization problem. An optimization upper bound is derived by means of Lagrange multipliers and KKT conditions. 3. The paper is well-written and organized.

Weaknesses: 1. The motivation of AE-KD is to encourage the optimization direction of the student guided equally by all the teachers. However, considering there are some weak teachers (low generalization accuracy) in the ensemble teacher pool, why are these weak teachers treated equally with other strong teachers in the gradient space? Intuitively, the guidance of student should favor those strong teachers, but keep away from the weak teachers. 2. The proposed AE-KD seems similar with the previous work OKDDip [1], where the dynamic attentions are learned by gradients to adaptively weight the teachers’ logits. What is the difference between them? 3. How to optimize the weights \alpha_m in Eq. (11)? Is it end-to-end optimized together with the student? If yes, how to ensure \alpha_m less than C during optimization? 4. The teacher resnet56 is trained for 240 epochs. Why is the student resnet20 trained for 350 epochs? 5. In ensemble learning, it is valuable that the ensemble networks have various architectures with different accuracies. How does AE-KD perform in this situation? [1] Online Knowledge Distillation with Diverse Peers. AAAI, 2020.

Correctness: The formulations are correct and easy to understand. But the motivation confuses me. Please refer to the above comments.

Clarity: The paper is well-written and organized.

Relation to Prior Work: The proposed AE-KD seems similar with the previous work OKDDip. The authors should discuss it and claim the differences between OKDDip and AE-KD.

Reproducibility: No

Additional Feedback: The author's feedback partially resolves my concerns. Finally, I decide to increase the overall score.


Review 3

Summary and Contributions: This work regards ensemble knowledge distillation as a multi-objective optimization problem and proposes a novel gradient-based algorithm, which computes a set of weights for different teachers based on gradients and decides how the student will learn from teachers in each minibatch. This method adaptively combines the knowledge from multiple teachers and thus provides the student with better guidance than using the traditional averaged loss.

Strengths: 1. This paper proposes a new viewpoint for ensemble knowledge distillation. 2. The method sounds reasonable and empirical results are promising.

Weaknesses: While the analysis in Section 2 is interesting, it is not so relevant to the main focus of this paper. It is better to remove this part. Several minor issues about experiments: 1. It is better to conduct multiple runs of experiments and report the mean and variance. 2. AEKD* is better than AEKD on CIFAR10 but worse on CIFAR100. Any explanations? 3. Figure 2 seems to suggest that with more teacher models, AVER will catch up the proposed method and even outperform it. I'd like to see the results with more teachers. 4. The teacher models are very weak and far from SoTA models. It might be better to conduct experiments with stronger teacher models. 5. Line 244 C\in [1:M]. Should it be C\in (1/M, 1)?

Correctness: Yes

Clarity: Yes

Relation to Prior Work: Yes

Reproducibility: Yes

Additional Feedback:

[Author Response · NeurIPS 2020]

**To Reviewer 1:** Thank you for your positive comments. We have investigated the effect of the disagreement parameter

$C$ in Section 4.3. Actually, we also did similar experiments on other teacher-student networks and ensemble sizes (20

and 25). The observations are also similar. We will include these results to improve the comprehensiveness of our

ablation studies. Also, we will cover more references to enrich the related work.

**To Reviewer 2:** Thank you for your detailed comments and suggestions.

**Re: motivation.** You may misunderstand our motivation. Our AE-KD actually **does not** treat every teacher in the

ensemble equally. On the contrary, we introduce a parameter $C$ to allow disagreement among teachers, so that the

obtained update direction of parameters is not necessarily a strict descent direction for all teachers. As you suggested,

the gradient directions from weak or noisy teachers are not as reliable as those from good teachers (also see line

161-166). With the help of parameter $C$, the final direction will not accommodate these weak teachers necessarily, and

a better weight over teachers can be automatically determined by solving Problem (9) or (11).

**Re: comparison with OKDDip.** The main differences lie in three ways. First, the learning paradigms are different.

OKDDip is essentially an online KD paradigm which adapts two-level distillation. While AE-KD follows traditional

teacher-student paradigm and the teachers won't be updated during the training. Next, we use different strategies to

learn the weights. In the second-level distillation of OKDDip, group knowledge is transferred to the group leader,

where all diverse peers serve as a group of teachers. The weights for diverse peers are computed using self-attention

mechanism. In AE-KD, the weights for teachers are computed based on multi-objective optimization in gradient space.

We take disagreement among teachers into consideration and prevent the student suffering from adverse guidance. Last,

AE-KD has the tunable parameter of disagreement to reconcile all teachers while OKDDip doesn't.

**Re: optimization of weights $\alpha_m$ in Eq. (11).** Eq. (11) is solved for every minibatch with the calculated gradients over

all teacher losses. It is a typical One-class SVM problem with constraint $0 \leq \alpha_m \leq C$, and can be easily solved by

LIBSVM or other off-the-shelf solvers.

**Re: 350 epochs for resnet20.** In experiments, we train all the teachers for standard 240 epochs, following the setting

in [1]. And for student resnet20, we train for 350 epochs for better performance.

**Re: ensemble networks with various architectures.** Our AE-KD performs in the same way, *i.e.*, every teacher

provides a gradient and the final direction is computed according to Eq.(9). To validate this case, we take resnet20

as student, and use resnet110, wide_resnet_40_2 and vgg13 as teacher networks with accuracy 74.31%, 75.61% and

74.64%, respectively. Results show that our AE-KD can achieve 70.16% accuracy while our baseline method AVER

only has 69.40%, showing our superiority of dealing ensemble KD even with various architectures. We will include this

setting in our final version.

**To Reviewer 4:** Thank you for your detailed comments and constructive suggestions for our experiments.

**Re: Section 2.** It intended to formally illustrate related work and baseline methods. We will consider your advice.

**Re: issues about experiments.**

**1: multiple runs.** Thanks for your suggestion. We experimented with five reset56 teacher networks and the resenet20

student network on CIFAR10, and run our AE-KD for 10 times with different random seeds. The mean and standard

variance are 92.49% and 0.02%, respectively. We can see the performance of student network tends to be steady (low

variance) due to the distillation from teacher networks. We will cover this in our final version.

**2: performance gap between AE-KD* and AE-KD on CIFAR10/100.** The performance gap comes from the weight

$\beta$ in Eq.(7) of the feature-based loss in AE-KD*. In real implementation, for CIFAR10 we determine the optimal $\beta$

by cross-validation in $\{10^{-1}, 1, 10, 100, 1000\}$, and for simplicity, we solely adopt the same $\beta$ for CIFAR100, thus the

performance of AE-KD* on CIFAR100 can drop a bit than AE-KD since we do not tune $\beta$ specially for CIFAR100.

However, what we emphasize here is that given a KD method (logits or feature based), when it comes to the ensemble

setting, our method can develop a better way to distill from their ensemble by adaptively assigning weights. Of course,

suitable parameters bring in better results. For example, if we tune $\beta$ specially for CIFAR100, our AE-KD* can have

71.95% accuracy on resnet20 student network with five resnet56 teacher networks, surpassing AE-KD (71.37%).

**3: experiments with more teachers.** With more teachers involved, the proportion of weak teachers will tend to be

steady. Thus for AVER, their influence on the performance will gradually reduce to a certain level, and the accuracy

will increase dramatically with more teachers at first, but stabilize eventually as Figure 2. However, our method remains

steady and robust for most time. We experiment with 30, 40, 50 teachers on CIFAR10. The performances of AVER and

AE-KD are 92.59%, 92.67%, 92.69% and 92.77%, 92.82%, 92.86%, respectively. AE-KD still outperforms AVER with

a comparable and stable gap for large ensemble size.

**4: weak teacher models.** Our teacher models follow those in [1] with the same or similar accuracy. Admittedly,

stronger teacher models do produce better students. We choose network from [2] on CIFAR10 as teacher models with

97.37% accuracy and the performance of AE-KD on resnet20 student network reaches 95.14% compared to AVER

(93.66%). The results are even better and AE-KD retains its superiority.

**5: line 244.** Yes. It's a typo. Thank you for pointing it out and we will fix it in our final version.

[1] Yonglong Tian, Dilip Krishnan, and Phillip Isola. Contrastive representation distillation. ICLR 2020.

[2] Hanxiao Liu, Karen Simonyan, and Yiming Yang. Darts: Differentiable architecture search. ICLR 2019.


[Meta-Review · NeurIPS 2020]

I recommend this paper for acceptance. This is a good idea and the research is well executed. All reviewers are also in agreement that this paper should be accepted.